# Membrane-Bound Meet Membraneless in Health and Disease

**DOI:** 10.3390/cells8091000

**Published:** 2019-08-29

**Authors:** Chujun Zhang, Catherine Rabouille

**Affiliations:** 1Hubrecht Institute of the Royal Netherlands Academy of Arts and Sciences, and University Medical Center Utrecht, 3584 CT Utrecht, The Netherlands; 2Department of Biomedical Science of Cells and Systems, University Medical Center Groningen, 9713 AV Groningen, The Netherlands

**Keywords:** membrane, organelles, phase separation

## Abstract

Membraneless organelles (MLOs) are defined as cellular structures that are not sealed by a lipidic membrane and are shown to form by phase separation. They exist in both the nucleus and the cytoplasm that is also heavily populated by numerous membrane-bound organelles. Even though the name membraneless suggests that MLOs are free of membrane, both membrane and factors regulating membrane trafficking steps are emerging as important components of MLO formation and function. As a result, we name them biocondensates. In this review, we examine the relationships between biocondensates and membrane. First, inhibition of membrane trafficking in the early secretory pathway leads to the formation of biocondensates (P-bodies and Sec bodies). In the same vein, stress granules have a complex relationship with the cyto-nuclear transport machinery. Second, membrane contributes to the regulated formation of phase separation in the cells and we will present examples including clustering at the plasma membrane and at the synapse. Finally, the whole cell appears to transit from an interphase phase-separated state to a mitotic diffuse state in a DYRK3 dependent manner. This firmly establishes a crosstalk between the two types of cell organization that will need to be further explored.

## 1. Introduction

Cell compartmentalization has been largely defined by the formation and maintenance of membrane-bound organelles, such as those displayed in the compartments of the secretory pathway, the endosomal-lysosomal pathway, the mitochondria, the peroxisomes, and the lipid droplets. Each sustains specific functions and biochemical reactions. However, this has recently been complexified by the discovery of membrane-contact sites between all these membrane-bound compartments through which lipids and ions traffic [1], as well as by the (re-)discovery of the so-called membraneless organelles (MLOs) [2].

MLOs define a class of compartments that are not membrane-bound, i.e., not surrounded by a sealed phospholipid membrane. They are formed by phase separation either liquid-liquid or liquid-solid [3]. Phase separation defines the behavior of a homogeneous solution of dispersed macromolecules that segregate into two distinct phases either liquid or solid/gel/fibrillar. The differences in the material properties of the separated phases can be distinguished by FRAP (Fluorescence Recovery After Photobleaching). In the case of liquid droplets, bleaching half of the structure would result in a quick recovery through the efficient movement of the non-bleached molecules to the bleached area within the droplet. This is the case for P granules (reviewed in [4]), P-bodies, stress granules [5] and Sec bodies [6]. When the structure is solid and crystalline, the recovery does not occur as the molecules within the structure are immobile. For instance, metabolic enzyme foci that form in starved yeast are solid [7]. Note that in this case, these material properties do not affect their reversibility. However, stress granules tend to transit from liquid droplets to solid entities, and this transition is accompanied by their irreversibility [5,8], a feature that is proposed to coincide with the onset of neurodegenerative diseases.

Phase separation is driven by “driver/scaffold” proteins [9] that are essential and sufficient to drive phase separation and formation of MLOs. When they are absent, these compartments are not formed or are not stable. Drivers coalesce and attract other proteins or macromolecules with which they normally interact. Drivers engage in mutivalency, i.e., in low affinity multivalent interactions [9]. They often contain domains of intrinsically disordered regions that contain repeating sequences or high frequency of the same amino-acid, such as proline, arginine, glutamine that can interact with other amino acids such as tyrosines [10]. The interactions involved in liquid phase separation are electrostatic but also cation/pi as well as pi/pi [11]. Overall, phase separation results in the formation of low affinity coalescence containing proteins and often (but not always) Ribonucleic acids (RNAs). These condensates are also referred to “quinary structures” in which the abundance of components is not entirely defined and may vary [12]. As such, they are different from the ribosome or the proteasome, which are complexes with a fixed subunit stoichiometry [13]. Last, as membrane-bound organelles, MLOs also sustain specific biochemistries. For instance, the RNA-binding protein Otu was shown to form MLOs that exhibit deubiquitination activity. Notably, this enzymatic activity only emerges when Otu coalesces, not as monomers [14]. Other examples of signaling regulated by MLOs is available in [15].

MLOs exist both in the cell nucleoplasm and in the cytoplasm, in basal steady state conditions (such as the nucleolus, nuclear speckles, centrosome, P granules in C.elegans [4], germ granules in Drosophila [16], the plant pyrenoid [17], and neuronal RNA granules [18]). MLOs also formed upon cellular stress, among which the best studied are the P-bodies and stress granules (recently reviewed in [19]).

However, the name “membraneless organelles” is misleading because, although they are not sealed by a membrane, they are not devoid of interaction with membrane. First, the formation of some MLOs is related to inhibition/modulation of specific membrane traffic steps in the early secretory pathway (Appendix A). They are also related to the cyto-nuclear transport (Appendix B). In this regard, some MLOs may incorporate proteins that normally function in these trafficking steps (part 1 and 2). Second, some MLOs form and are located near a membrane, and membrane can enhance their coalescence (part 3). Last, the entire cell is now proposed to transit from a phase-separated state in interphase to a dispersed state in mitosis (part 4).

Here, we will review what is known of the link/interface between MLOs, membrane, membrane traffic, and its components. However, because this review will illustrate that the term MLOs is not appropriate, we will not use it further (although we keep in in the title). Instead we will use the term “phase-separated biocondensates” when they occur in basal conditions and “stress assemblies” when they are triggered by cellular stress.

## 2. Stress Assemblies and the Inhibition of Trafficking in the Early Secretory Pathway

### 2.1. P-Bodies Form in Yeast Mutant for Secretory Pathway Components

P-bodies are submicron dynamic cytoplasmic stress assemblies comprising translationally inactive mRNAs and proteins that are normally associated with translation repression, mRNA turnover such as 3′-deadenylation, 5′-decapping, 5′-3′ exonuclease activity, nonsense mediated decay, and miRNA-targeted gene silencing in all eukaryotic cells so far examined [20]. Mammalian P-bodies are characterized by the presence of AGO1/3, DCP2, EDC3 among many other proteins [21,22] whereas Dcp1p, Dcp2p, Edc3p mark the yeast P-bodies [23]. In yeast, P-bodies are only visible where they are stressed for instance by oxidative stress, endoplasmic reticulum (ER) stress, osmotic shock, glucose starvation [24,25]. In mammalian cells, P-bodies are visible as small microscopic entities even in the absence of stress (reflecting RNA degradation needed for cell homeostasis), but stress triggers their enlargement [26].

Given their concentration in RNA decay factors, P-bodies have been proposed to be the site of mRNA degradation and turnover, but it appears that they are more involved in RNA storage without degradation [21,27]. The contradictory presence of intact mRNAs and RNA decay factors in P-bodies is puzzling but may reflect protection of the mRNAs by specific RNA-binding proteins and translational repressors that inhibit degradation.

With their proposed role in RNA biology, P-bodies are not intuitively linked to membrane and membrane traffic. Yet, a first link has been made with the discovery that in the yeast thermosensitive mutant *arf1-11* [28], the P-body number increases even at normal growth temperature, a phenotype that is referred to, as the “multiple P-body phenotype” [29]. This phenotype is even more pronounced upon stress by heat shock [29]. Arf1 (ADP-ribosylation factor 1) is a small GTPase that plays an important role in vesicular trafficking especially in the Golgi where it appears characteristically concentrated. Indeed, Arf1 activation is the first step in the assembly Coatomer complex I coat (COPI) that mediates the retrograde transport from the Golgi to the ER [30]. At the Golgi, Arf1 also modulates the activity of phospholipase D [31]. In addition, Arf1 is proposed to have novel and conserved roles in the morphological and functional maintenance of mitochondria [32,33].

As mentioned above, P-bodies are hardly visible in wildtype yeast in growing conditions, but their number increased in growing *arf1* mutants. This increase is strikingly strong because neither starvation nor oxidative/redox stress leads to such an increase in wildtype yeast. Interestingly, the multiple P-body phenotype is not specific for *arf1* mutant. It is also observed in other secretory mutants [29], suggesting that this phenotype is most likely related to a general defect in secretion. Of note, it is unrelated to the activation of the unfolded protein response [29].

In yeast, the secretory pathway is massively used for the transport and secretion of the components of the cell wall that is important to protect cells from osmotic shock. Accordingly, *arf1* and other secretory yeast mutants are more sensitive to osmotic stress. When treated with high salt, P-body formation increased further. However, this is specific to salt and not to glycerol, suggesting that this phenotype is not downstream of osmotic stress per se, but is related to salt stress.

Salt stress can lead to a transient increase in intracellular calcium [34,35]. Accordingly, adding CaCl_2_ (but not MgCl_2_) to the medium induces the multiple P-body phenotype in wildtype cells. Furthermore, treatment of *arf1* mutants with the calcium chelating agent BAPTA (1,2-*bis*(*o*-aminophenoxy)ethane-N,N,N′,N′-tetraacetic acid) results in the reduction of the number of P-bodies that form in this mutant. Taken together, the multiple P-body phenotype observed in secretory mutants is largely dependent on a change in intracellular calcium. In line with it, Calmodulin, a major player in calcium signaling is required for this phenotype. However, it is required only for the multiple P-body phenotype observed in secretory mutants, not when this phenotype is induced by other stresses [29] (Figure 1A).

This suggests that P-body formation can be induced by differential signaling pathways. Specific P-body components could therefore be required for P-body formation upon one stress but not another. In this regard, Pat1 (that contains an EF-hand, which might coordinate a calcium) and Scd6 (an Sm-like protein most likely involved in the regulation of mRNA translation and/or degradation in PBs [36]) are both required for P-body formation in secretory mutants, but do not play a role in their formation upon glucose starvation [29].

How the inhibition of the early secretory pathway triggers calcium imbalance in the cytoplasm that leads to the formation of P-bodies (related to RNA mutant homeostasis) remain to be further investigated, especially if it does not trigger ER stress. This opens interesting avenues to integrate membrane traffic and the formation of phase-separated biocondensates.

### 2.2. Sec Body Formation and the ER Exit Sites (ERES)

In the previous section, yeast mutants inhibiting the function of the early secretory pathway displayed an increase in their cytoplasmic calcium resulting in P-body formation. Here, we address how nutrient stress modulates the early secretory pathway (Appendix A) leading to the phase separation of ERES components into a different organelle, the stress assembly named Sec bodies.

Amino-acid starvation of Drosophila S2 cells leads to inhibition of protein transport out of the ER [6]. This is also the case in mammalian cells, at least when measured with the Sec-luciferase reporter [37]. In S2 cells, this stress also leads to the coalescence/condensation of Sec bodies [6]. Although Sec body composition is still not firmly established despite the analysis by mass spectrometry [38], they definitely contain ERES components, Sec16, and COPII subunits Sec23, Sec24, and Sec31. In other words, they contain components related to the trafficking step that is inhibited. Of note, Sec bodies, unlike P-bodies are not RNA-based.

Sec bodies have a diameter between 0.6–0.8 μm and they were shown by electron microscopy to not be sealed by a membrane although they are often close to ER. This agrees with the fact that they form at the ER cup marking the ERES in non-stressed cells. Two lines of evidence suggest this. First, live cell imaging reveals that out of the 15–20 ERES in growing cells, only a couple are “selected” and will form Sec bodies, while the other disappear. The model is that the selected ERES will be remodeled into nascent Sec bodies while the ERES components of the other ERES are released and recaptured by the growing Sec bodies. Second, marking the ER cup with a form of Sec16 that only contains the domain that targets it to ERES [39] but that is not incorporated in Sec bodies [6], clearly showed that Sec bodies form very close to the ER cup, even though they can later be displaced. This suggests that Sec bodies initially form where ERES were present.

Sec bodies have the material properties compatible with being a liquid droplet. FRAP experiment of a small part of the structure showed a relatively fast recovery, emphasizing that this condensate is not an aggregate. Sec bodies also reversible very quickly upon amino-acid addition. Sec16 and COPII subunits form functional ERES upon addition of amino-acids after the starvation and they also appear protected against degradation during the starvation period [6]. Last, Sec bodies are pro-survival during the period of stress and upon stress relief as many nutrient stress assemblies are (reviewed in [19]).

Given that they incorporate COPII coat subunits (although not Sar1, and not COPI coat components), the role of trafficking in the early secretory pathway in their formation has been investigated. However, neither Sar1 depletion that leads to a reduction in the COPII coated vesicle formation nor treatment with Brefeldin A (BFA) that blocks COPI coated vesicle formation affects Sec body formation. This supports the notion that that protein transport in COPI and COPII coated vesicles is not necessary for Sec body formation [6].

What is? As mentioned in the introduction, drivers in the phase separation into liquid droplets are known to be components displaying low complexity sequences with intrinsically disordered domains, such as those found in RNA-binding proteins (see part 2). Interestingly, the ERES components mentioned above, Sec16 and the two Sec24 isoforms in Drosophila (Sec24AB and CD) display a higher content of low complexity sequences when compared to the overall Drosophila proteome [6]. Accordingly, depletion of Sec24AB (but not of Sec24 CD) [6] and Sec16 [40] reduces Sec body formation. The low complexity sequences of Sec24AB are clustered at its N-terminus. When those are tagged with the Green Fluorescent Protein (GFP), they can recruit GFP to Sec bodies, suggesting that they are sufficient to mediate this recruitment. For Sec16, the situation is more complex as only the central conserved domain of the protein is folded, the rest being largely intrinsically disordered. A domain of 140 amino acids at the C-terminus of the protein has been shown to be required for Sec body formation [6]. Within this region, a smaller conserved domain of 44 amino acids (that we named SRDC for Serum Responsive Domain Conserved) appears to play a major role in Sec body formation. First, a full-length version of Sec16 deleted of this domain does not support Sec body formation. Second, overexpression of SRDC in the absence of stress leads to Sec body formation even if it is not itself recruited to Sec bodies. [40]. The mechanism behind this finding is still lacking. One tentative explanation is that this peptide displaces Sec16 from the ERES by competing its interaction with COPII subunits. This is however unlikely as Sec16 without this domain still localizes to ERES [40] and that overexpression of full-length Sec16 does not lead to ectopic Sec body formation (unpublished). We propose instead that upon stress Sec16 changes slightly its conformation and exposes its SRDC that becomes a clear cis acting factor in Sec body formation. Overexpression of this domain could somehow mimic or induce this change in conformation.

The change in conformation could be triggered by post-translation modifications. In this regard, Drosophila PARP16 appears to play a key role in Sec body formation. Human PARP16 is a mono ADP-ribosylation (MARylation) enzyme [41,42] suggesting that dPARP16 has the same activity even though the catalytic site of the Drosophila enzyme is not entirely conserved. Depletion of dPARP16 prevents their formation and its overexpression in growing cells leads to their ectopic formation. Critically, the depletion of dPARP16 in S2 cells increased the sensitivity to amino-acid starvation and the cells do not recover after stress relief. Interestingly, dPARP16 appears to MARylate SRCD, adding a large pi contribution to its interaction. Overexpression of SRDC in dPARP16 depleted cells does not result in Sec body formation [40]. This suggested that the MARylation of SRDC in a dPARP16 dependent manner is necessary and sufficient for Sec body formation. How does amino-acid starvation activate dPARP16 remains to be established.

Taken together, these two examples show that inhibition of the early secretory pathway leads to the formation of stress assemblies. Whether the pathways are the same in both cases needs to be further defined. For instance, it will be interesting to also establish if calcium plays a key role in Sec body formation as it does for P-bodies.

What is remarkable in the case of Sec bodies is that the driver for their formation is a protein (Sec16) that organizes the step that is inhibited (optimal formation of COPII coated vesicles at ERES) [43]. The formation of Sec bodies in turn acts as a protection again Sec16 (and COPII subunits) degradation. This opens the possibility that inhibition or activation of membrane traffic steps (for instance in the endocytic pathway) triggers the formation of other condensates to be discovered. Why not imagine that inhibition of endocytosis leads to clathrin-based condensate that would also serve as a protective reservoir?

## 3. The Complex Relationship between Stress Granules, Cyto-Nuclear Transport, and Amyotrophic Lateral Sclerosis (ALS)

### 3.1. Stress Granules

Stress granules are also submicron stress assemblies that form in eukaryotic cells upon many cellular stresses as long as they induce inhibition of translation initiation and polysome disassembly. This leads to an accumulation of untranslated, 80S ribosome-free mRNAs in the cytoplasm that can bind RNA-binding proteins that phase separate into membraneless compartments, the stress granules [44]. They often form adjacent to, or overlapping with, P-bodies, yet their two functions appear distinct. Stress granules have been proposed to act as triage centers for mRNAs [45] protecting capped and polyadenylated mRNAs from degradation in P-bodies [46]. Stress granules would then store these protected mRNAs in such a way that they can be immediately translated upon stress relief [46,47,48].

As Sec bodies, Stress granules are rapidly reversible upon stress relief and they are pro-survival. Indeed, cells that do not form stress granules survive less well during stress and thrive less upon stress relief (recently reviewed in [19]).

### 3.2. Stress Granules and ALS: A Role for FUS, TDP43, and C9orf72

There is a solid relationship between stress granules and ALS (and frontotemporal dementia). ALS is a disease characterized by the degeneration of motor neurons and ultimately neuronal death leading to muscle weakness. Genetically, ALS can be linked to mutations in RNA-binding proteins, such as Ataxin2, Fused in Sarcoma (FUS), TAR DNA-binding protein 43 (TDP43), and C9orf72 (the latter accounting for 40% of familial forms of ALS).

The pathological hallmark of ALS is the presence of inclusion bodies (i.e., abnormal aggregations of proteins) in the cytoplasm of motor neurons. In about 97% of people with ALS, the main component of the inclusion bodies is TDP-43 protein [49] except in the case of FUS mutations where the main component of these inclusions is FUS itself [49,50].

FUS and TPD43 are both RNA-binding proteins containing prion-like domains that are intrinsically disordered and prone to “coalescence”. For wildtype purified FUS [5,51,52] and TDP43 in vitro, this coalescence corresponds to a phase separation into liquid droplets [5,53,54]. This is driven either by cation-pi interactions including those between tyrosine residues present in the prion-like domain and arginine present in the RNA-binding domain [55] or through LARK (low-complexity, aromatic-rich, kinked segments) interactions [56] (reviewed in [3]). In cells, wildtype FUS localizes to stress granules that are formed upon oxidative stress (such as arsenite treatment) at least in mammalian cultured cells [5].

Conversely, mutated FUS and TDP43 in vitro also phase separate, but in assemblies that become increasingly solid, fibrillar, and irreversible with time. In vivo, when mutated FUS and TDP43 proteins are expressed at the same time as stress granules form (for instance upon oxidative stress that is shown to probably be, at least partly, at the origin of the disease onset), they affect their dynamics. Stress granules are no longer liquid and no longer reversible.

There are two slightly different models explaining how mutated FUS and TPD43 mutations exert their deleterious effect on stress granules dynamics and material properties. Mutated FUS is still recruited to stress granules (as wildtype FUS), but it starts forming irreversible aggregates within the granules leading to modifications of their properties [57] (Figure 2B). TDP43 acts differently (Figure 2B). The recruitment of wildtype TDP43 to stress granules (that is mediated by a Tankyrase dependent mechanism [58]) is proposed to prevent TDP43 phosphorylation. It appears that TDP43 phosphorylation is a key event leading to its pathological aggregation. In this regard, mutated TDP43 that is not recruited to stress granules is therefore prone to phosphorylation and consequently aggregate outside stress granules [58]. Despite the lack of overlap, stress granule dynamic is nevertheless affected, perhaps because TDP43 aggregates interfere with their ability to regulate the RNA targeting to them, or through signaling [59] (Figure 2B).

The mutation of *C9orf72* causing ALS consists on a hexanucleotide GGGGCC repeat expansion [60]. In healthy individuals, this hexanucleotide is typically present less than 20–30 times [61] but in ALS patients, it can be repeated more than a few hundred times [62]. There are several theories about how *C9orf72* GGGGCC repeat expansion causes ALS. It appears that the RNA transcribed from the *C9orf72* gene containing the expansion is translated through a non-ATG initiated mechanism leading to the synthesis and accumulation of dipeptide repeat proteins that can affect cellular homeostasis in multiple ways. One of them is that this accumulation compromises cyto-nuclear transport [63,64,65]. The second is that this dipeptide repeat proteins themselves phase separate/aggregates and impair stress-granule dynamics in such a way that they also become irreversible [66].

The relationship between stress granules and cyto-nuclear transport is complex. Cyto-nuclear transport is mediated by nuclear transport receptors, including nuclear import receptors, NIRs that import cognate proteins to the nucleus (Appendix B). Certain NIRs do localize to stress granules and consequently, stress granule formation inhibits cyto-nuclear transport (Figure 2A). Conversely, NIRs can act as chaperones for RNA-binding proteins containing prion-like domains (such as FUS and TDP43) and prevents their deleterious aggregation that affect stress granule dynamics (Figure 2B). Below we review both bodies of evidences.

### 3.3. Stress Granules Assembly Negatively Regulates Cytoplasm to Nucleus Import

The first observation that components of nuclear transport machinery localize to membraneless organelles was made in C. elegans. There, the nucleoporin NUP98 that localizes on the nucleoplasmic face of nuclear pore complexes also associates to the large cytoplasmic membraneless RNA-based P granules [67]. This observation was extended to stress granules where several nuclear transport receptors were shown to assemble to stress granules in mammalian HeLa cells treated by arsenite, diethyl maleate as well as in cells subjected to heat stress [68]. Interestingly, P-bodies are not targeted 

If indeed, nuclear transport receptors are present in, and recruited to, stress granules under conditions of cellular stress, it might ultimately inhibit the nuclear trafficking through the nuclear pore. This is exactly what was found for the NIR Karyopherin β2 (also called importin β2 and transportin 1) that also localizes to stress granules in arsenite treated cells [69]. This recruitment is concomitant with the strong inhibition of nuclear import as shown by using a Shuttle-tdTomato (with both a nuclear localization signal (NLS) and nuclear export signal (NES) flacking the tomato dDNA) that was found accumulating in the cytoplasm [69] (Figure 2A).

Furthermore, as stress-granule dynamics have been linked to ALS pathology triggered by mutated *C9orf72* and *TDP43*, the role of these mutations on nuclear transport was assessed. Accordingly, expression of the C9orf72 Dipeptide repeat proteins [63,64,65] and mutated TDP43 [70] disrupt nucleocytoplasmic transport. Critically, the inhibition in nuclear transport is due to the formation of impaired stress granules that contain NIRs, such as Karyopherin β2. Consequently, inhibiting stress granule assembly, for instance by depleting Ataxin 2, suppresses nucleocytoplasmic transport defects as well as neurodegeneration in C9ORF72-mediated ALS and Frontotemporal dementia [69].

### 3.4. Beyond Nuclear Transport: NIRs Act as Chaperones to Prevent Stress-Granule-Related Pathological Aggregation in ALS (Figure 2B)

As summarized above, certain nuclear transport receptors can be recruited to stress granules, the formation of which affects cyto-nuclear trafficking. What is also clearly shown is that NIRs has other functions that are independent of their role in nuclear import. Indeed, they act as cytoplasmic chaperones and promote the physiological dynamics of stress granules. Briefly, when FUS and TDP43 are bound to their cognate NIRs, they can enter the nucleus, and the pool that is present in the cytoplasm is dispersed. When FUS and TDP43 can no longer bind their cognate NIRs (either because there are mutated or the level of NIRs is lower or mutated), they form irreversible aggregates that interfere with stress granule function, dynamics, and reversibility leading to the proteinopathy observed in ALS. Experimental evidences underlying this exciting new biology has been gathered both in vitro and in vivo by three independent groups in back-to-back articles [53,71,72] and we outline them below.

#### 3.4.1. Purified FUS Phase Separation into Liquid Droplets is Specifically Inhibited by Karyopherin β2

As reported above, purified FUS and HnRNPA1 phase separate in vitro into liquid droplets. Strikingly, addition of their cognate purified NIRs (i.e., Karyopherin β2) [73] efficiently prevent their phase separation [53,71,72]. Furthermore, when added to preformed FUS and hnRNPA1 liquid droplets, Karyopherin β2 also rapidly disperses them [53].

FUS phase separation is however not affected by adding NIRs that do not bind FUS, such as importin 5 [72] and Impα/β [71]. Kap121, a NIR that binds weakly to FUS only partially prevents FUS phase separation [71]. This suggests that when NIRs have a low affinity for a cargo, they are not able to prevent their phase separation in vitro. Accordingly, FUS chimera harboring an NLS that is recognized by Impα/β phase separates and addition of purified Impα/β then prevents it.

In vitro, many different RNA binding proteins such as FUS (as well as hnRNPA1, hnRNPA2, TAF15 and EWSR1,) can be made to fibrillate, and addition of purified Karyopherin β2 strongly inhibits this fibrillization. As above, addition of either a mutant Karyopherin β2 that cannot bind its cargoes or the complex Impα/β do not alter FUS fibrillation [53]. Importantly, the effect of NIRs to chaperone RNA binding proteins and prevent their aggregation is not limited to Karyopherin β2. Indeed, the fibrillization of TDP43 (TDP43^Q331K^) that is a cognate substrate for Impα/β is prevented by addition of this specific NIR. As predicted, Karyopherin β2 does not prevent seeded TDP43 fibrillization.

Taken together, the presence of Karyopherin β2 and impα/β strongly reduces the in vitro phase separation/fibrillation of FUS and TDP43, respectively (Figure 2B).

#### 3.4.2. Does Karyopherin β2 Antagonize the Coalescence/Aggregation of FUS in Cellulo?

Human FUS overexpressed in yeast mis-localizes to the cytoplasm and aggregates, as in neurodegenerative diseases [74]. However, co-expression of human Karyopherin β2 prevents the formation of these aggregated foci and allows FUS to localize to the nucleus. In addition, FUS^P525L^ and FUS^R495X^ that cannot bind Karyopherin β2 remain aggregated and are not affected by the expression of Karyopherin β2, validating the in vitro results. [53].

In human stressed cells, endogenous FUS is recruited to stress granules upon cellular stress and elevating Karyopherin β2 expression inhibits this accumulation. This is independent from the role of Karyopherin β2 in nuclear import because Karyopherin β2 expression has the same effect on wildtype FUS and a FUS version deleted of its PY-NLS, the binding motif to Karyopherin β2 that is critical for nuclear import [53].

Additional evidence is provided by mammalian cells expressing a version of FUS that is not imported in the nucleus. It is maintained in a diffused state in cytoplasm because of the presence of Karyopherin β2. When EGF-M9M (a peptide that inhibits Karyopherin β2 binding to its cargoes) is co-expressed, FUS localizes into stress granules. This further indicates that FUS binding to Karyopherin β2 prevents its recruitment to stress granules and therefore prevents its deleterious effect on their dynamics. Accordingly, when Karyopherin β2 is not present or prevented to bind FUS, FUS is efficient recruited to stress granules (Figure 2B). Conversely, in cells expressing EGFP-biMax (a peptide that inhibit the binding of importin to its cargoes), cytoplasmic FUS remains diffuse because it does not affect its binding to Karyopherin β2 [72]. This was further tested in a semi intact cell system in which stress granules are first formed, followed by the cell permeabilization that washes out endogenous Karyopherin β2 and blocking the nuclear pores. When FUS is then added with purified Karyopherin β2, its recruitment to stress granules is significantly reduced [72] (Figure 2B).

Taken together, Karyopherin β2 reduces FUS accumulation and aggregation in stress granules, thus preventing them to become dysfunctional and irreversible.

#### 3.4.3. Where Does Karyopherin β2 Bind FUS to Prevent Their Condensation/Fibrilization?

Karyopherin β2 binds stably the PY-NLS of FUS and this allows FUS import to the nucleus. Interesting, PY-NLS is also critical for Karyopherin β2 to prevent FUS phase separation in vitro. As in cells (see above), the ability of Karyopherin β2 to prevent FUS phase separation is abolished by adding the inhibitory peptide M9M [71]. However, the binding domain of Karyopherin β2 to FUS is not restricted to the NLS and extends to other domains. In depth Nuclear Magnetic Resonance spectroscopy analysis reveals that Karyopherin β2 directly binds the RGG2 and RGG3 of FUS [71]. Karyopherin β2 binding to the FUS RGG3-PY was confirmed independently [72]. Importantly, this purified arginine rich domain (RGG3-PY) phase separates on its own, in agreement with [75], and addition of Karyopherin β2 suppresses it. Strikingly, both phase separation and binding to Karyopherin β2 is inhibited when these arginines are mutated to lysines [72].

The role of RGG3 in FUS phase separation properties was also examined in the context of full-length FUS. Indeed, deletion of the PY-NLS does not affect FUS liquid phase separation but mutating all arginine residues in RGG2 and RGG3 of full-length FUS to lysines strongly decreases phase separation, demonstrating a clear role of the RGG3 domain [71]. In conclusion, Karyopherin β2 chaperones FUS through its binding the RGG3-PY domain to prevent its condensation

#### 3.4.4. How Does This Relate to ALS and FTD? Could NIR be Used as A Treatment of ALS?

FUS^R495X^ and FUS^P525L^ are two ALS-linked FUS variants which can cause aggressive juvenile ALS, and these two mutations reduce FUS binding to Karyopherin β2 [72,73]. In vivo Karyopherin β2 reduced FUS^P525L^ aggregation by ~50% while having limited activity against FUS^R495X^ [53]. In vitro, both wildtype and mutated proteins phase separate but Karyopherin β2 only reduces the condensation of wildtype FUS, not this of mutant proteins [72]. Thus, the presence of these mutations has similar effect as reducing the level of Karyopherin β2, and results in irreversible aggregation.

As mentioned above, Karyopherin β2 can rapidly dissolve preformed FUS fibrils and hydrogel in vitro. In parallel, Impα/β also leads to the disaggregation of TDP-43 and TDP-43^Q331K^ fibrils. The effect of Karyopherin β2 in reverting pathological aggregates was tested in a Drosophila model for neurodegeneration. Elevating Karyopherin β2 rescues FUS-Linked Neurodegeneration and partially rescues the lifespan of the flies [53]. This parallels the finding that elevated Karyopherin β2 also mitigates FUS^R521H^-mediated motor neuron degeneration in Drosophila [76]. This indicates that NIRs could be used to prevent prion-like spread of self-seeding aggregates during ALS as well as other neurodegenerative diseases [77].

Taken together, Karyopherin β2 appears to behave in two different manners regarding stress granules: On one hand, it is incorporated to stress granules resulting in the inhibition of cyto-nuclear import that is a hallmark of ALS. On the other hand, it acts as cytoplasmic chaperone for ALS related FUS, thus preventing its recruitment to stress granules. Note that Impα/β similarly chaperones TDP43. In this case, TDP43 is prevented to aggregate next to stress granules and exert its deleterious effect at a distance. In both cases, the interaction between stress granules and NIRs is at the center of this disease, but in in two opposite manners. In the first case, stress granule formation appears to promote the disease as they recruit NIRs and contribute to the inhibition of cyto-nuclear transport observed in C9orf72. In the second case, stress granules are protected by NIRs (that do not appear to localize with them) as they chaperone FUS and TDP43 and prevent these proteins to affect the dynamics and reversibility of stress granules. More work is of course needed to sort out the role of other NIRs.

## 4. Membrane Enhances the Formation of Phase-Separated Condensates

### 4.1. P-Body Formation is Regulated by ER Proteins

As mentioned above in the introduction, the term “membraneless” is not completely appropriate to define biological condensates that arise from phase separation in vivo. For instance, P-bodies that form in yeast are often localized in close proximity to ER membranes [29] in line with what has been observed in the oocyte Drosophila bodies that form around the ER membrane [78].

To identify ER proteins mediating the ER localization of P-bodies in yeast, and more generally to shed light on the relationship between ER and P-bodies, a deletion screen targeted to ER proteins was conducted in wildtype yeast in which P-bodies (marked by Dcp2 and Scd6) were induced upon heat shock. Bfr1 and Scp160 were identified as modulating P-body formation, but not their localization near the ER [79]. Scp160 is an mRNA-binding protein that, together with Bfr1 (whose function is still unclear), associates with actively translating polysomes at the ER [80]. They both localize to the ER membrane whether the yeast are stressed or not, and both Scp160 and Bfr1 are shown by immunoprecipitations to interact with P-body components (such as Dcp2) [79] (Figure 3A).

However, the phenotype observed in Bfr1 or Scp160 deletions is complex. There are more P-bodies in these mutants when compared to wildtype, but only in basal conditions. When the growth temperature is upshifted, the mutant cells have less P-bodies than wildtype. Therefore, Scp160/Bfr1 negatively regulate P-body formation under normal stress-free growth conditions [79].

### 4.2. Plasma Membrane Receptors Promotes Phase Separation

Not only stress assemblies form in contact with membrane but phase separation can also occur and be enhanced directly at biological membranes through transmembrane proteins. At the plasma membrane, this would lead to efficient receptors clustering together with their cytoplasmic interactors and potentially their ligands. Furthermore, the proximity of a membrane (a 2D flat lipid bilayer) could enhance phase separation perhaps by concentrating components and facilitating their pi–pi and cation/pi interaction.

As briefly reported in the introduction, specific interactions between protein domains can drive phase separation. In vitro reconstitution with purified components have shown that purified SH3 domains and proline-rich motifs support phase separation into liquid droplets, even though their affinity for each other is not very high [81]. In vivo, the ability of these domains to phase separate is used to create domains on the plasma membrane that enhance enzymatic reactions. This is the case for the polymerization of actin cytoskeleton around the transmembrane adhesion receptor Nephrin. Indeed, Nck and N-WASP are two cytoplasmic SH2 and SH3 domains containing proteins that are involved in the formation of actin filaments. Through its SH2 domain, Nck first associates with three phosphorylated tyrosines present in the long disordered cytoplasmic tail of Nephrin [82]. Once bound, the three SH3 domains of Nck binds the proline-rich domain of N-WASP, an event that drives phase separation in the plane of the plasma membrane and leads to the clustering of Nephrin. As Nephrin binds the Arp2/3 complex, it becomes activated and efficiently drives the polymerization of actin (Figure 3B).

Similar clustering and condensation of SH3- and SH2- domains containing plasma membrane adapters has also been reported in the contact of T cell receptor activation mediated by the T cell receptor complex protein LAT (linker for activation of T cells). In this case, LAT coalescence attracts specific proteins (such as ZAP70) and repulse others such as CD45. This would then create a micro-environment where specific, spatially restricted and efficient T cell receptor signaling reactions takes place. Altogether, this leads to an amplification of T cell signaling [83,84]. Membrane clustering therefore appears to be a common mechanism in cellular signaling, which may serve to stabilize active conformations, amplify signals and may introduce switch-like behavior [15].

### 4.3. The Synapse, A Phase Separation Mediated by Small Lipid Carriers?

As reported above, a flat plasma membrane can help promoting phase separation in acting as a platform or a crucible. However, small lipid carriers can also enhance phase separation as observed in an in vitro system mimicking the synapse [85].

Synapsin1 is a protein that is present at the synapse. It harbors an ATP (Adenosine triphosphate) binding module that is flanked by an intrinsically disordered C-terminal regions that are, in principle, prone to separate into liquid phase. To test this, in vitro purified GFP-tagged synapsin 1 forms micrometer-size droplets that tend to fuse together. Furthermore, FRAP of a small area within the droplet reveals that the phase is liquid and that Synapsin1 moves in and out of the droplet rapidly. Analysis of two purified fragments of Synapsin1 confirmed that the intrinsically disorder region is responsible for droplet formation, not its folded central ATP binding module [85].

As mentioned above, SH2 and SH3 domains containing proteins interact with intrinsically proline-rich domains containing proteins, such as synapsin 1. Interestingly, two SH3 domains containing proteins normally present at the synapse are intersectin (an endocytic multidomain scaffold protein and an regulator of synaptic vesicle replenishment) [86] and Grb2 [87] (growth factor receptor–bound 2) and they both interact with Synapsin1 via its proline-rich intrinsically disorder tail. When tested in vitro, incubation of purified Synapsin1 with intersectin and Grb2 leads to their co-phase separation into larger droplets. The use of the sole SH3 domain of intersectin yields a similar effect, supporting as above, the role of SH3 domain and its interaction with proline-rich domain in phase separation.

Synapsin1 is normally present at the synapse that is, a crowded environment. Accordingly, the phase separation propensity of purified Synapsin1 (alone and in the presence of its SH3 binding partners) is enhanced by presence the crowding agent Polyethylene glycol (PEG). Given that SH3 containing proteins and PEG play a positive role in Synapsin1 phase separation, and given that at the synapse, Synapsin1, intersectin and Grb2 are all present at the cytoplasmic leaflet of synaptic vesicles [88], this infers that Synapsin1 phase separates at the synapse in a manner that depends on its interaction with SH3 domain proteins and on crowding. 

However, the presence of small lipidic synaptic vesicles could also play a role in this phase separation. This was tested in vitro. Small artificial 50–150 nm diameter lipid vesicles devoid of proteins (but labeled by the lipid dye Cy5-DOPE) were added to purified synapsin1. Strikingly, the formation of synapsin liquid droplets correlated with the accumulation of lipid dye in the droplets. No droplets were observed in the absence of synapsin 1 and the vesicles do not condensate on their own. Analysis by electron microscopy showed that these droplets are clusters of small vesicles connected by synapsin 1, while in the absence of synapsin 1, vesicles remained dispersed. Importantly, synapsin 1 condensates did not recruit vesicles lacking negatively charged phospholipid suggesting that those are necessary for synapsin1 binding to vesicles even in the absence of Grb2 and intersectin [85].

These results show that the clustering of small lipid carriers and synapsin 1 might mimic the synapse in a minimalistic manner. Indeed, observed a few decades ago by EM, synapses are characterized by a collection of clustered synaptic vesicles seemingly glued together by a protein gel. This gel is likely to contain synapsin 1 [89]. Of course, at the synapse, the vesicles would also harbor SH3 containing proteins at their surface and in addition to lipids, this will likely help the phase separation of synapsin 1. The next step of the in vitro assay will be to mix purified synapsin 1 together with small lipidic carriers carrying SH3 domains (Figure 3C).

Taken together, these findings opens the possibility to study the two forms of cell compartmentalization (membrane-bound and non-membrane-bound) in an integrated manner, refining further how membrane drives phase separation and what molecular contact are established between these two classes of organelles, in the same way as membrane contacts sites are now studied. For instance, are the ER anchored P-bodies moving together with the ER? Is there an exchange of components, including protein and RNAs between condensates/stress assemblies and membrane-bound organelles? Furthermore, as membrane enhances the formation of condensates, it also facilitates biochemical reactions taken place in these condensates. What are those reactions and how are they different from those that occur in more dispersed state remains to be investigated? Finally, it will be important to understand how biological condensates influence membrane dynamics, membrane traffic, and the function of membrane-bound organelles in general.

## 5. The Cell Cycle Appears to be A Global Transition from An Interphasic Phase-Separated State to A Mitotic Dispersed State

Last, the concept of phase transition has recently been extended to the whole cell during the cell cycle. What is proposed is that the cell in interphase is characterized by the presence of many MLOs, which become diffuse at the onset of mitosis in a DYRK3 dependent manner [90]. This is proposed to ensure optimal partitioning in the two daughter cells, a phenomenon that is reminiscent of membrane-bound organelles that fragment at the onset of mitosis. DYRK3 is a kinase that has been identified previously as being required for the dissolution of stress granules upon stress relief. This DYRK3 dependent dissolution links to TORC1 signaling as it releases the TOR kinase that was trapped in the granule during stress [91].

To understand the role of DYRK3 in non-stressed cells, DYRK3 was overexpressed which resulted in its spontaneous phase separation that occurs at a specific concentration threshold. At low expression level in interphase cells, DYRK3 shows a diffuse distribution within the nucleus and the cytoplasm with some enrichment at centrosomes and in nuclear splicing speckles. Interestingly, at high expression, it leads to the dissolution of these splicing speckles, an effect that was reverted by treating cells with the DYRK3 inhibitor. Furthermore, when DYRK3 is prevented to enter the nucleus, the splicing speckels are no longer dissolved. Overexpression of DYRK3 also dissolves pericentriolar satellites and prevents the induction of stress granules with arsenite in the cytoplasm in a kinase-activity-dependent manner. However, nucleoli of overexpressed DYRK3 cells were still intact. In mitotic cells, overexpressed DYRK3 localizes to spindle poles and biochemically, DYRK3 interacts with proteins that are heavily phosphorylated during mitosis [90].

Therefore, what is the role of DYRK3 when the content of both cellular compartments mixes at the onset of mitosis? The first clue is that inhibition of DYRK3 in mitotic cells leads to the aberrant condensation of the splicing-speckle marker SC35, the stress-granule marker PABP (PolyA-binding protein) and the pericentriolar material protein PCM1. They were found to co-condense into a so-called liquid phase-separated hybrid mitotic granules that also contain for polyadenylated RNA and accumulated inhibited DYRK3. However, these structures are not positive for P-body, Cajal body markers, and ubiquitin. It therefore appears that DYRK3 kinase activity is essential during mitosis to prevent the formation of liquid-liquid phase-separated mitotic granules [90].

Overall, overexpression of DYRK3 in interphase cells leads to the dissolution of many MLOs (but not the nucleolus) while inhibition of DYRK3 in mitosis leads to the formation of a non-physiological hybrid granule. DYRK3 is therefore proposed to act as a dissolvase of multiple liquid-unmixed/phase separated/condensate compartments to prevent their aberrant condensation during mitosis.

How would the same enzyme have a low activity in interphase and be activated at the onset of mitosis? DYRK3 expression increases gradually as cells progress from late S to the end of G2 (mitosis) with sudden increase in DYRK3 levels as cells enter mitosis. Accordingly, the DYRK3-to-substrate ratio has been proposed to modulate the phase transition. The notion is that when the ratio is high, the phase separates should disperse (like during mitosis). Conversely, when this ratio is low, condensation should take place (like in interphase). To investigate this, non-mitotic cells were transfected with inducible DYRK3 and SRRM1 (a protein a splicing-speckle with a high content in low complexity regions), and the changes in their concentration was recorded by time-lapse imaging, from G1 to mitosis exit. As expected, the DYRK3/SRRM1 ratio increases over time and SRRM1 remains dissolved in cells that display a high DTRK3/SRRM1 ratio. This indicated that this ratio is required to maintain the dissolved phase during mitosis. Conversely, when the ratio is low, SRRM1 is present in granules, as observed in interphase.

In conclusion, interphase cells appear to exhibit a low level of DYRK3 which allows the phase separation of many physiological MLOs. Conversely, mitotic cells have a higher level of DYRK3 as well as many mitotic phosphor-proteins that allow the dissolution of these MLOs.

## 6. Conclusions

In this review, we have reported that the two forms of cell compartmentalization, membrane-bound organelles and MLOs, interact with one another more than was anticipated. Inhibition of membrane traffic steps by mutations or cellular stress leads to the formation of specific stress assemblies, some of them based upon proteins that are key for optimizing on of this step (such as Sec16, part 1). We also report how stress granules interact in a complex manner with nuclear transport receptors and as such are involved in ALS (part 2). In the third part, we report that although the two forms of cell compartmentalization (membrane-bound and membraneless) are often studied independently of one another by different groups of researchers, they in fact interact. Even better, membrane enhances phase separation of condensates (part 3). Last, it appears that the whole cell transits from a phase-separated state at interphase to a more dispersed state at the onset of mitosis, an observation that is reminiscent to what happens to membrane-bound compartments that fragment and disperse at mitosis. The notion behind this is that it would enhance the chances of equal partitioning between the two daughter cells even if we now know that it is in fact not stochastic.

Altogether, the link between membrane-bound organelles and phase-separated condensates has several consequences. First, although the term “membraneless organelle” is language-wise correct when defined as non-sealed by a membrane, it also alludes to the notion that phase separation in cellulo “has nothing to do” with membrane. Consequently, the term membraneless organelles should been replaced by “phase-separated condensates” or “stress assemblies” as we have done in this review. Second, the contact between these condensates/assemblies and membrane needs to be defined in a more systematic manner as they influence each other’s biology. P-bodies appear to form near the ER and what about other condensates given that the ER pervades the entire cell? Third, a large amount of data leading to the deep understanding of the biophysical and material properties of these condensates are performed in vitro using purifying proteins. However, given that the cell is an environment crowded with membrane-bound organelles that interact with condensates, now is a good time to add membrane to in vitro phase separation experiments. In this regard, the pioneering experiments performed with synapsin 1 and small lipid carriers in vitro are paving this avenue to recreate in a small (and still reductionist) manner the cell interior.

## Figures and Tables

**Figure 1 cells-08-01000-f001:**
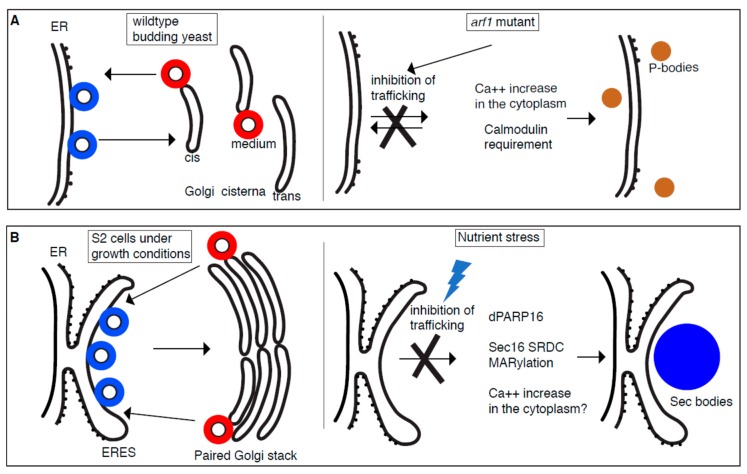
Stress assemblies and the inhibition of trafficking in the early secretory pathway. In growing conditions, COPII (Coatomer complex II) coated vesicles (blue) bud from ER (budding yeast) and ER Exit Sites (ERES) (Drosophila S2 cells) whereas COPI (Coatomer complex I) coated vesicles (red) bud from the Golgi. (**A**): In yeast, a subset of secretory transport mutants causes inhibition of trafficking in the early secretory pathway and leads to the formation of multiple P-bodies. These mutants are more sensitive to salt leading to an increase of calcium in the cytoplasm and triggering the formation of specific P-bodies containing Pat1 and Scd6 in a calmodulin dependent manner. (**B**): In Drosophila S2 cells, the nutrient stress of amino acid starvation also inhibits protein transport in the early secretory pathway and leads to the formation of Sec bodies. Amino-acid starvation appears to lead to dPARP16 activation that MARylates the Sec16 SRDC (Serum Responsive Domain Conserved) leading to Sec body formation. Whether Ca^++^ imbalance is also important remains to be established.

**Figure 2 cells-08-01000-f002:**
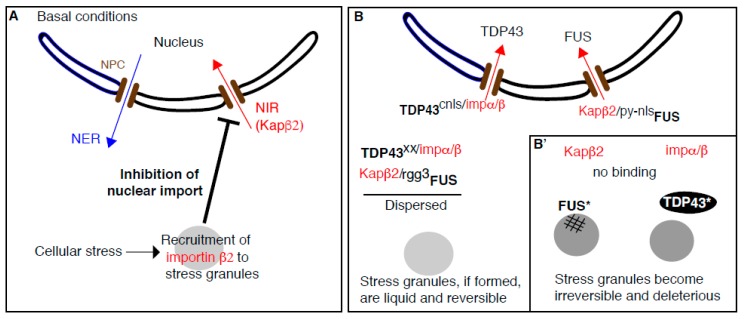
The complex relationship between stress granules, cyto-nuclear transport and ALS. (**A**): In basal growth conditions, the nuclear import receptor (NIR) Karyopherin β2 (Kapβ2) associates to its cognate cargoes and imports them to the nucleus through the nuclear pore complex (brown bars on the nuclear envelope). Upon cellular stress, Kapβ2 is recruited to stress granules and is no longer available to function as a receptor. As a result, stress granule formation triggered by cellular stress causes inhibition of cyto-nuclear transport. (**B**): The RNA-binding protein FUS and TDP43 are normally imported in the nucleus by binding to their cognate NIRs (Kapβ2 and Impα/β, respectively). The pool of FUS and TDP43 that is present in the cytoplasm remains dispersed through their binding to Kapβ2 and Impα/β, respectively but via other domains (such as FUS-rgg3). (**B’**): Upon cellular stress and/or expression of mutated FUS and TDP43 that prevent their binding to their NIRs, mutated FUS (FUS*) and TDP43 (TDP43*) aggregates in a deleterious manner but in two different ways. FUS* is recruited to stress granules and aggregates within, leading stress granules to be become solid and irreversible. On the other hand, TDP43* aggregates on its own next to stress granules and negatively impact their dynamics via a mechanism that is still to be elucidated.

**Figure 3 cells-08-01000-f003:**
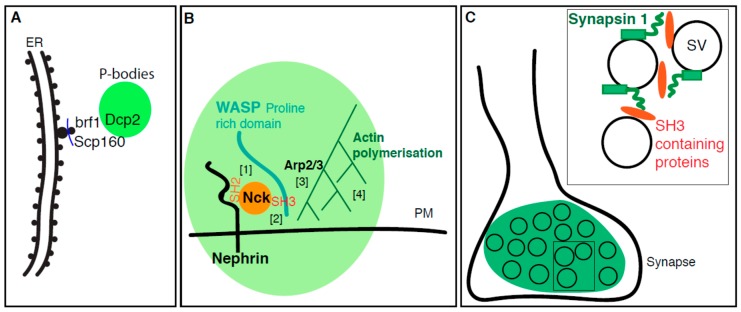
Membrane enhances the formation of phase-separated condensates. (**A**): P-bodies form near the ER as DCP2 interacts with the ER proteins Scp160 and Brf1 themselves interacting with translating ribosomes present at the translocon. (**B**): Nephrin is a transmembrane adhesion receptor resident of the plasma membrane (PM). It binds to Nck-SH2 domain via its long disorder cytoplasmic tail [1]. Nck contains also three SH3 domain can binds the proline-rich domain of N-WASP [2]. This forms a biocondensation (green) that recruits Arp2/3 [3], resulting in actin polymerization around Nephrin [4] (**C**): Synapsin 1 at synapse interacts with several SH3 domain containing proteins (in orange such as Intersectin and Gbr2) via its intrinsically disordered region. These interactions are proposed to lead to the phase separation of these components and coalescence of synaptic vesicles (SV) as observed at the synapse.

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
