# Peer review of "Membrane-Bound Meet Membraneless in Health and Disease"

_cells, 2019, doi:10.3390/cells8091000_

Round 1

Reviewer 1 Report

Membrane-less organelles (MLO) recently came to the forefront of cell biology studies. The review by Zhang and Rabouille focuses on the insufficiently studied aspect of MLO biology – their interactions with cell membranes. The authors discuss the interplay between MLO and ER/Golgi under normal conditions and stress, the relationships between MLO formation/dissolution and cytoplasmic-nuclear transport, the influence of ER and plasma membranes and their protein components on MLO formation and, finally, they address the induction of MLO dissolution by DYRK3 kinase during mitosis. This is a well written review devoted to a biologically important problem that has significant medical implications (e.g. role of MLOs in amyotrophic lateral sclerosis). The analysis of the existing literature is very comprehensive. Some small oversights must be corrected:

In the line 75 “submicromolar” needs to be changed to “submicron”. FTD needs to be spelled (frontotemporal degeneration).

Author Response

We wish to thank the reviewers and editor for their reading and commenting our review. We have taken their comments at heart and addressed them in full. We have also tried to minimize the number of typos and missing words.
Our changes are marked in brown in the main text.

1. “P-bodies are submicromolar dynamic”. Replace submicromolar by submicron
>>This has been done.

Reviewer 2 Report

Often it looks like the authors did not even take care in re-reading the draft: there are several incomplete sentences, punctuation missing, sentences confusing and clearly not properly checked.

Moreover, the description of some detailed findings from the literature needs a good Figure to refer to (in addition/substitution to the quite vague Fig. 1 presented in the manuscript, as one example); instead the authors often give for granted that all readers that may be interested in this topic, know the molecular details of what the authors are describing. This makes the review much less clear and informative/useful than it could be.

If the authors will seriously consider rewriting the manuscript, I am willing to act as referee, since as I already mentioned before, I believe the topic is of interest and timely.

Author Response

We have corrected all the mistakes and misleading sentences that impede clarity. We hope that the reviewer will be satisfied.